# Machine Learning, Urban Water Resources Management and Operating Policy

**Evangelos Rozos**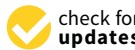

National Observatory of Athens, Institute for Environmental Research & Sustainable Development, GR-15236 Athens, Greece; erozos@noa.gr

**Abstract:** Meticulously analyzing all contemporaneous conditions and available options before taking operations decisions regarding the management of the urban water resources is a necessary step owing to water scarcity. More often than not, this analysis is challenging because of the uncertainty regarding inflows to the system. The most common approach to account for this uncertainty is to combine the Bayesian decision theory with the dynamic programming optimization method. However, dynamic programming is plagued by the curse of dimensionality, that is, the complexity of the method is proportional to the number of discretized possible system states raised to the power of the number of reservoirs. Furthermore, classical statistics does not consistently represent the stochastic structure of the inflows (see persistence). To avoid these problems, this study will employ an appropriate stochastic model to produce synthetic time-series with long-term persistence, optimize the system employing a network flow programming modelling, and use the optimization results for training a feedforward neural network (FFN). This trained FFN alone can serve as a decision support tool that describes not only reservoir releases but also how to operate the entire water supply system. This methodology is applied in a simplified representation of the Athens water supply system, and the results suggest that the FFN is capable of successfully operating the system according to a predefined operating policy.

**Keywords:** machine learning; water resources management; optimization; stochastic modelling

---

## 1. Introduction

Water resource management is a notoriously difficult problem, mainly because of the uncertainty regarding inflows to the system, thereby making inevitable the probabilistic approach [1], which accordingly sets a framework of policy decisions related to the acceptable risk of failure [2]. To support the implementation of policy decisions, both classical and data-driven methodologies have been employed.

Machine learning has been emerging as a very efficient multi-functional tool in all scientific fields. Besides the information technology sector, machine learning is finding its way into drug discovery, agriculture, and even the legal industry [3]. Conversely, it has been almost three decades since artificial intelligence (AI) applications (genetic algorithms, fuzzy logic, neural networks, etc.) appeared in hydrology. Back in 1992, French et al. [4] attempted to forecast rainfall (short-term) employing a neural network with three layers. In water resources management, Raman and Chandramouli [5] employed a feedforward neural network (FFN) to formulate an operating rule for a single reservoir for irrigation. Since then, FFN applications have evolved with increasing complexity and frequency. For example, Chandramouli et al. [6] trained an FFN to estimate optimum releases from a system of three reservoirs. The data required to train this FFN was obtained from a dynamic programming algorithm that simulated the system over a period of 36 years with a simulation time step of two weeks. Cancelliere et al. [7] combined a soil-water balance model, dynamic programming, and a neural network to derive the

operating rule of a reservoir, which supplies water for irrigation. Other researchers have employed more synthetic approaches that put FFN together with various state-of-the-art (of that time) tools. For example, Chang et al. [8] used an adaptive network-based fuzzy inference system to estimate the optimal water release from a single reservoir in Taiwan. To train the FFN, they combined the knowledge of the operating rules in effect with the optimal reservoir operating histogram, obtained from the application of a genetic algorithm. Their study showed that this approach achieved a much better performance than the traditional one.

Despite their considerable ingenuity, all the previous applications were found to have two main drawbacks. The first one is that the water system is not simulated as a single entity. The storage elements are simulated by water balance equations, whereas the transmission elements, if simulated at all, are by some external hydraulic model. For example, Mansouri et al. [9] employed dynamic programming to optimize the design of four dams and three water transfer tunnels; the tunnels were simulated with EPANET. Though more complicated systems can be studied following this methodology, there is currently no software available to help with all the required procedures. This means that model integration, network definition, discretization of states and actions, etc., should be made either manually or with custom-made software. The second drawback has to do with the 'curse of dimensionality' of the dynamic programming method [10]. According to Castelletti et al. [11], the computational effort for optimizing $n$ reservoirs for $T$ time steps is proportional to $T\,(N_s{}^n\,N_d{}^n)$, where $N_s$ and $N_d$ are the number of elements in the discretized state, and release decision sets respectively. Furthermore, if the inflow uncertainty is considered, the resulting Bayesian stochastic dynamic programming [12] carries an even higher computational cost, since the number of the state variables increases by the number of inflows and forecasts [13]. Chaves and Kojiri [14] tried to address this issue by employing stochastic fuzzy neural networks, which intrinsically consider the stochastic nature of the problem and can be trained directly, without dynamic programming, to minimize operating costs. This method, though much faster than the two-step optimization (first the application of the dynamic programming and then the training of the neural network), is still compute-intensive for two reasons. First, an extra loop within the training process is introduced to account for the stochastic variable—the inflow. Second, the genetic algorithm, which is used for optimization, is much slower than any gradient-based optimization method, which is used in FFN training.

An additional problem with the previous methods is training FFN only on historical data. If the period happens to exclusively include either only dry years or only wet years, the conditional probabilities of inflows (the probability of an inflow when knowing the previous inflow, i.e., prior flow transition probabilities in Bayesian stochastic dynamic programming) will not correctly represent the stochastic properties of the system. Consequently, the resulting operating rule will not meet the requirements of the operating policy (e.g., operate the system with a higher than the anticipated risk of failure).

To avoid the previously mentioned issues, a different approach is suggested in this study. The concept is similar to the methodology proposed by Lobbrecht and Solomatine [15], who used neural networks to replace a computationally intensive conventional controller with a fast 'intelligent controller' to control flood prevention in Delfland. These intelligent controllers allow the deployment and delivery of tailored tools that are more intuitive to stakeholder. For example, the tool can be implemented as a spreadsheet or a GIS based application that helps a utility manager to plan the operation of a water supply system. Furthermore, researchers can use this approach to achieve any level of model coupling [16] of decision support models with other types of models.

## 2. Materials and Methods

The suggested methodologiy comprises three steps (see Figure 1):

1. Instead of dynamic programming, network flow programming (NFP) is used for optimizing the water supply system. In NFP, a water supply system is represented with a graph of $N$ nodes (e.g., points where two aqueducts join, reservoirs, etc.) and $A \subset N \times N$ links between the nodes

(including the carry-over artificial arcs that simulate the storage elements). The computational effort of an efficient NFP algorithm, e.g., RELAX4, is proportional to $T N \operatorname{card}(A) \log(N C)$, where $C$ is the range of fluctuation of the penalty functions' values [17]. Therefore, it is feasible to optimize multi-reservoir water supply systems over a continuous state space. Furthermore, NFP simultaneously performs simulation and optimization of the whole water supply system, and there are programs that offer friendly GUI and CAD environments to define the topology of the water supply system [18].

2. To successfully manage water resources, it is important to properly analyze the stochastic structure of the reservoir inflows. Bras and Rodriguez-Iturbe [19] have highlighted that the autocovariance estimator employed in classical statistics underestimates the autocovariance terms with large lags. This can underestimate the duration of the droughts. To overcome this issue, Koutsoyiannis suggested an approach combining a generalized autocovariance function with a coupling of stochastic models of different time scales [20,21]. This approach is capable of reproducing the persistence of multivariate processes (i.e., '*the tendency of annual average streamflows to stay above or below their mean value for long periods*' [20]), essential for reliably estimating the long-term fluctuations of the reservoir inflows.

3. The typical output of multi-reservoir management tools is the abstraction or releases from the available water resources. However, this output does not directly assist in system operations. The releases/abstractions need to be routed to the demand locations. This may be straightforward in the case of simple water supply systems (e.g., single reservoir and a single-line aqueduct); however, in the case of more complicated systems, a model may be required. To avoid this additional effort, a demand-oriented approach is employed in this study (see UWOT pull-signals in [22]). In this approach, the generation and routing of the water demand are simulated, instead of the flows. Instead of training the FFN to approximate the releases/abstractions, this approach trains it to approximate the ratio of the flows wherever two or more aqueducts join. With these ratios available at each time step for the entire water supply system, the calculation of releases/abstractions is simply a matter of multiplying the incoming demand signal by the appropriate sequence of ratios on the path from the demand point to each resource.

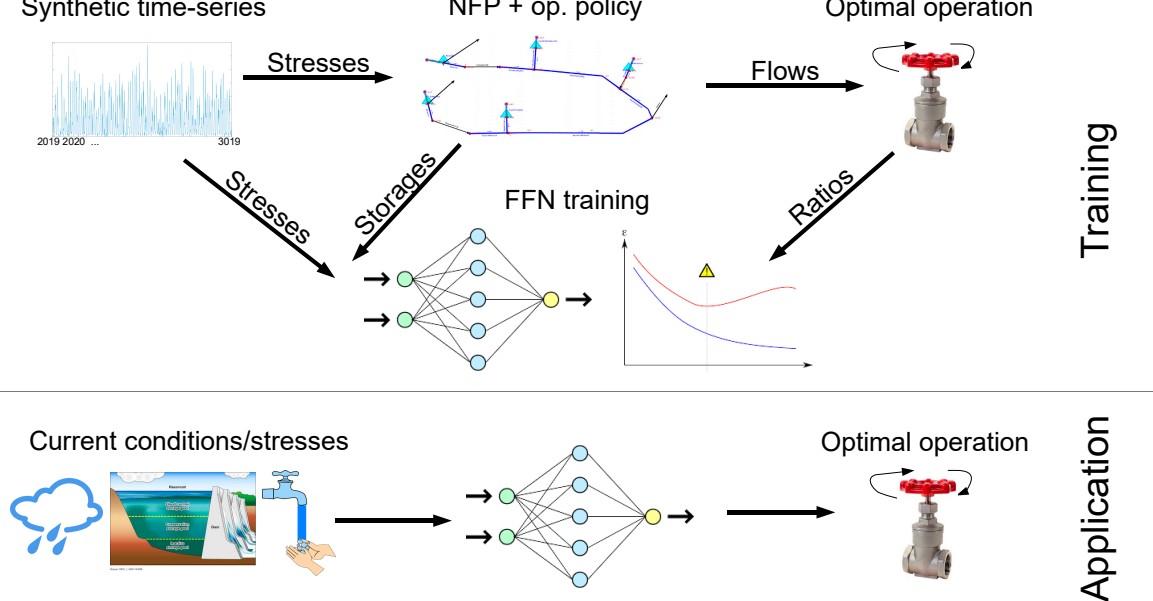

**Figure 1.** Schematic representation of the suggested methodology.

## 3. Case Study

The suggested methodology is demonstrated in the Athens water supply system. However, the same procedure can be repeated at any other case study. The water supply system of Athens has two major reservoirs (Mornos and Hyliki), one smaller nearby reservoir (Marathon), and two locations with boreholes where groundwater can be abstracted from the underlying aquifers. The water flows from the Mornos reservoir to Athens by gravity, whereas water pumps are required to abstract water from Hyliki. The capacity of the aqueduct that brings water from the Mornos reservoir suffices to cover the water demand of Athens. On the contrary, all the other resources together cannot provide enough water to meet the water demand of Athens. For this reason, whenever the Mornos reservoir gets empty, the system fails. Consequently, to increase the reliability of the system, water should be obtained from the energy-intensive resources when Mornos reaches a critically low level. This level depends on the acceptable risk, a policy decision made by the water utility company.

For the purposes of this study, a simplified conceptualization of the Athens water supply system was employed. This network was optimized with MODSIM 8.5.1. MODSIM was selected because it is the only free NFP tool [18] that simulates the water allocation in river basins through a sequential solution of the network flow optimization problem for each time period [23]. On the contrary, HEC-ResPRM, another free NFP tool, minimizes the objective function for all time steps and elements simultaneously [24]. The latter implies that the solution suggested at each time step is optimal in respect of all previous and following stresses on the system. In operational applications, the stresses following the current system state are in the future. A stochastic approach could resolve this issue, but this would significantly increase the complexity, as has been mentioned previously.

Figure 2 displays the representation of the simplified Athens water supply network. The nodes labelled with a name starting with "Rnf_" introduce time-series of runoff to the network, i.e., the inflows to the reservoirs. The nodes labelled with a name starting with "Cnf_" correspond to the locations where aqueducts join. The nodes labelled with a name starting with "Dmd_" correspond to the locations of water demand. The nodes labelled with a name starting with "GW_" correspond to the locations of groundwater abstractions. The nodes labelled with a name starting with "Rsv_" represent the reservoirs of the water supply system. Finally, the nodes labelled with a name starting with "Spill_" simulate the overflows from the reservoirs. The arrows linking two nodes represent the aqueducts.

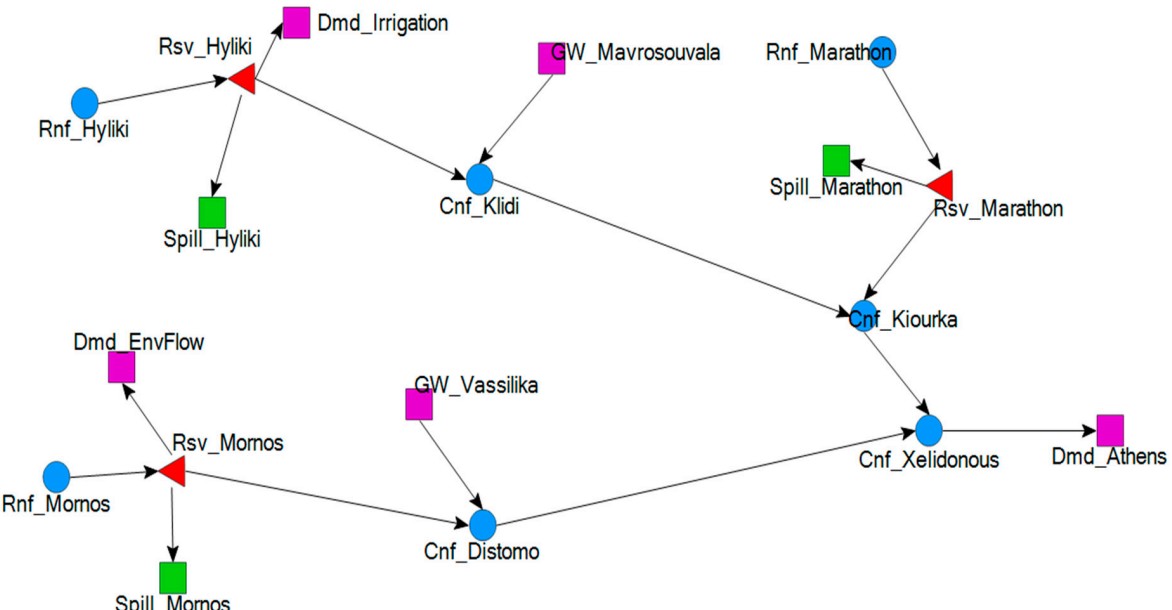

**Figure 2.** Representation of the simplified Athens water supply network in MODSIM.

The inputs to the water supply system are the inflows to the three reservoirs, whereas the outputs cover the demand for Athens water supply, the demand for irrigation, and the demand for environmental flow. Historical monthly data (time-series of inflows and water supplied to Athens) are available from 1 January 1996 to 1 December 2010 (180 months). Monthly synthetic data was generated by employing the Castalia stochastic model [25]. The synthetic inflow time-series produced by Castalia start on 31 October 2019 and end on 31 October 3019 (12,000 months). Synthetic time-series of water demand was produced by repeating the historical pattern.

The solution suggested by MODSIM minimizes the objective function (see Equation (1) of [23]), which is a combination of penalty functions for the water availability in the reservoirs and the energy consumption of the water pumps. The water availability penalty function is set equal to $-10,000\,f_{it}$ for Mornos and Hyliki (priority 4000, see Equation (4) of [23]), and $-10,010\,f_{it}$ for Marathon (priority 3999), where $fit$ is the storage in the reservoir $i$ during the time step $t$. In this study, 4 alternative operating policies were assessed to verify the applicability of the suggested methodology. These were implemented employing 4 different sets of energy consumption penalty functions (Table 1) assigned to the appropriate links.

**Table 1.** Operating policies (OP) as implemented with the penalty values assigned to the appropriate network links.

| OP | Link Hy-Kl [1] | Link Ma-Kl [1] | Link Va-Di [1] |
|----|----------------|----------------|----------------|
| 1 | 45 | 150 | 23 |
| 2 | 19,565 | 65,217 | 10,000 |
| 3 | 45 | ∞ | ∞ |
| 4 | ∞ | ∞ | ∞ |

[1] Link Hy-Kl is between Rsv_Hyliki and Cnf_Klidi, link Ma-Kl is between GW_Mavrosouvala and Cnf_Klidi, link Va-Di is between GW_Vassilika and Cnf_Distomo, see Figure 2.

Operating policy No. 1 imposes the lowest penalty to energy consumption and has the lowest risk of failure. Operating policy No. 4 imposes the highest penalty to the energy consumption to deter the usage of energy-intensive resources, but has the highest risk of failure. The values given to these penalties are selected to achieve the desired cost/risk balance and to preserve the ratio between the corresponding specific energies (e.g., the energy required for pumping water from the Hyliki reservoir is around two times the energy required in Vassilika groundwater abstractions). The annual probability of failure is estimated with the following formula (see Equation (4.26), [26]):

$$R = 1 - \exp(-12/\theta), \tag{1}$$

where 12 is the number of months of a year, $\theta$ is the mean number of months between failures, which is calculated dividing 12,000 (the length of synthetic time-series in this specific case study) by the number of times the Mornos reservoir gets empty during the simulation period. The value calculated by Equation (1) should not exceed the acceptable risk set by the water utility company.

The NFP optimizations with the synthetic stresses and the energy penalties of Table 1 gave 4 sets of time-series of reservoir storages and water flows between nodes. Four FFN were trained, one for each set of energy penalties, to serve as a substitute of the NFP model. These 4 FFN have identical topology. Only the values of the FFN inputs/outputs are different.

The inputs to the FFN were the storages of the three reservoirs, the inflows to the three reservoirs, the demand for supplying Athens with water, the demand for irrigation, and the month number (a number rotating between 1 and 12) of each simulation step (9 inputs in total). The outputs were the ratio of the flows mixing to the 4 locations where aqueducts join (Cnf_Klidi, Cnf_Kiourka, Cnf_Distomo, and Cnf_Xelidonous, see Figure 2).

The FFN hidden layer includes 21 nodes, and the activation function is the hyperbolic tangent sigmoid. The output layer contains 4 nodes, and the activation function is linear (Figure 3). Both inputs and outputs are normalized to have zero mean and a standard deviation of one.

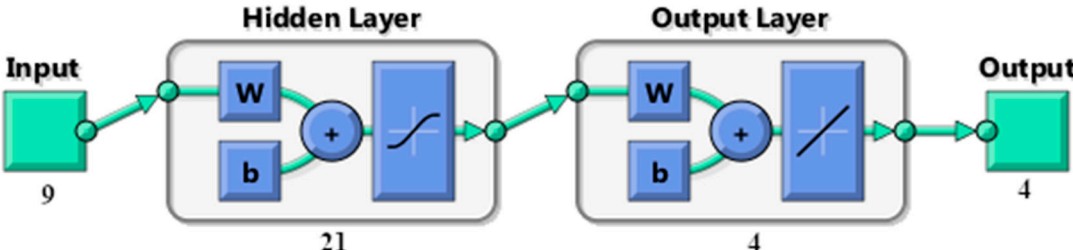

**Figure 3.** Topology of the feedforward neural network (FFN).

The last 2000 out from the 12,000 records of the synthetic data were used for validation. The Levenberg-Marquardt algorithm [27] was used to train the network with performance metric the mean squared error (MSE). To avoid overfitting, the network training was kept running while the performance continued to improve on the validation data set, but stopped once that performance started getting worse (early stopping, see [3]). The FFN was set up in Octave 5.1 [28] with the package nnet.

After training the 4 FFN with the synthetic data, they were ready for suggesting the optimum operation of the system. In real applications, only one FFN needs to be trained, which will encapsulate the experience of the optimal operation of the network for the predefined operating policy. The outputs of the FFN were the ratios of flows to the 4 locations where aqueducts join. Since at most two aqueducts join at all 4 "Cnf_" nodes, a single number suffices to describe the ratio of the incoming flows to these 4 nodes, hence the 4 FFN outputs. The FFN with the topology displayed in Figure 3 is equivalent to the formula:

$$O = f_{\text{out}} (f_{\text{hid}}(J \, W_{\text{hid}} + b_{\text{hid}}) \, W_{\text{out}} + b_{\text{out}}), \tag{2}$$

where $O$ (with dimensions $1 \times 4$ for the topology of Figure 3 and for a single time step) is the neural network output, $f_{\text{out}}$ and $f_{\text{hid}}$ are the activation functions of the output and hidden layers respectively, $J$ ($1 \times 9$) is the FFN input, $W_{\text{hid}}$ ($9 \times 21$) are the weights of the hidden layer, $b_{\text{hid}}$ is the vector with the bias terms ($1 \times 21$) of the hidden layer, $W_{\text{out}}$ ($21 \times 4$) are the weights of the output layer, and $b_{\text{out}}$ ($1 \times 4$) is the vector with the bias terms of the output layer.

In this study, UWOT [18,29] was employed to route the water demand according to the ratios given by Equation (2) and calculate the water budget of the reservoirs. To test the performance of the four FFN under realistic conditions, they were applied using the historical time-series.

## 4. Results

The training performances of the 4 FFNs that correspond to operating policies No. 1 to 4 are given in Table 2. Only the correlation coefficient of the output with the worst performance is reported. On the contrary, the MSE metric includes all FFN outputs (outputs give percent rates ranging from 0 to 100%). All performances refer to the validation period (2000 months of synthetic data).

**Table 2.** Training performances of the FFN under the 4 operating policies (OP).

|  | OP1 | OP2 | OP3 | OP4 |
|---|---|---|---|---|
| Correlation coefficient | 0.92 | 0.91 | 0.95 | 0.95 |
| MSE | 30.1 | 19.1 | 24.7 | 23.7 |

The annual probabilities of failure estimated from Equation (1) were 0% for No. 1 and 2, and 2% for No. 3 and 4 operating policies respectively (storages of the Mornos reservoir were obtained from the MODSIM simulations with the synthetic time-series).

Figure 4 displays the abstractions from the Mornos reservoir for the 4 operating policies during the historical period. In operating policy No. 1 the abstraction is reduced to the minimum possible to preserve the storage in this reservoir during the driest period of the historical data. Abstraction is significantly reduced in operating policy No. 3 also, but to a lesser extent.

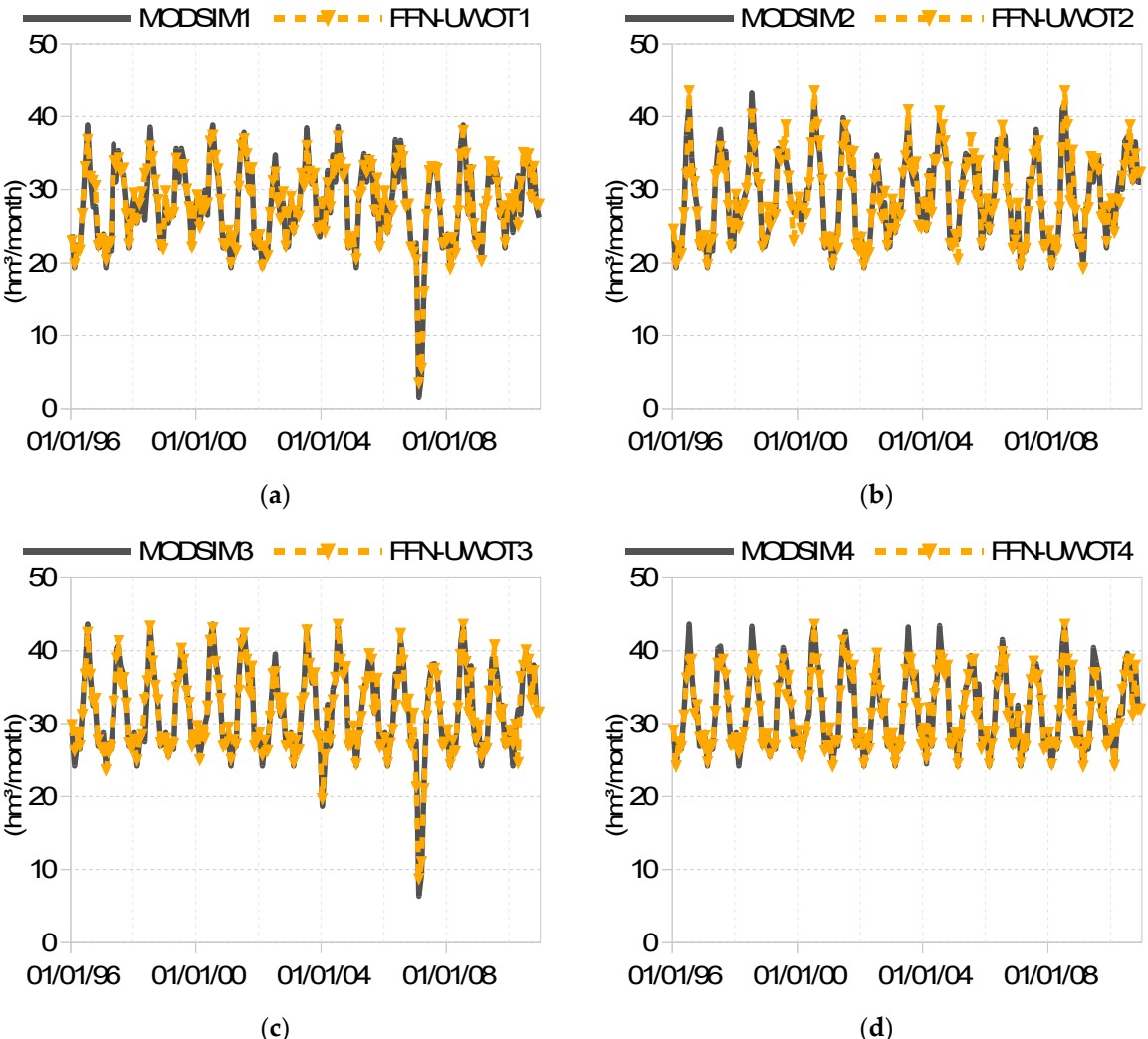

**Figure 4.** Abstractions from the Mornos reservoir obtained with MODSIM, and FFN with UWOT for operating policies No. 1 (**a**), No. 2 (**b**), No. 3 (**c**), and No. 4 (**d**).

Figure 5 displays the abstractions from the Hyliki reservoir for operating policies No. 1 and 3 during the historical period. Only operating policies No. 1 and 3 allow abstractions of water when the storage in Mornos drops below a certain threshold and, at the same time, the inflows are reduced. The other two operating policies do not use any water from the Hyliki reservoir.

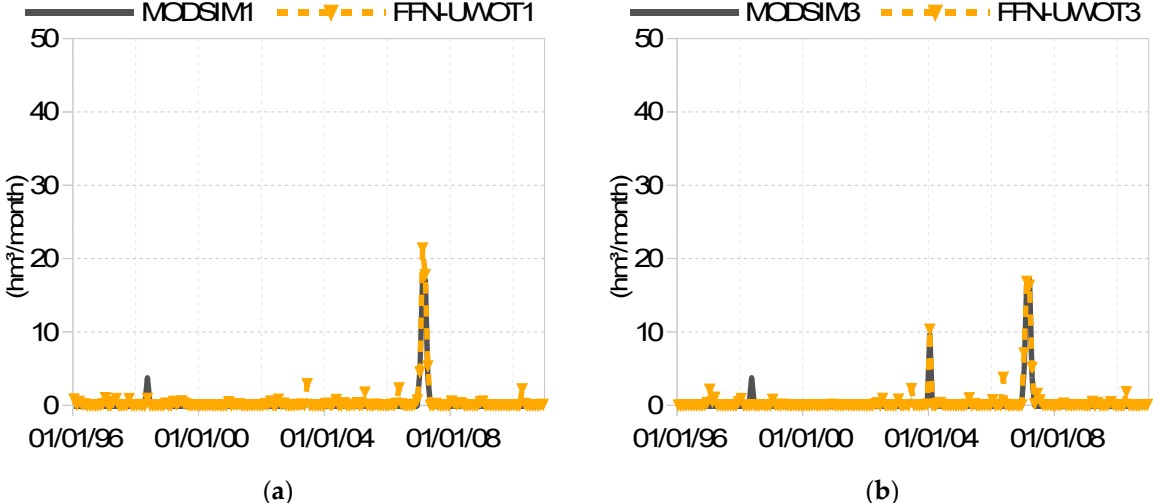

**Figure 5.** Abstractions from the Hyliki reservoir obtained with MODSIM, and FFN with UWOT for operating policies No. 1 (**a**) and 3 (**b**).

Figure 6 displays the abstractions from the Vassilika boreholes for operating policies No. 1 and 2 during the historical period. Water abstraction is allowed only in these two operating policies, with intermissions whenever the storage in Mornos reaches its maximum capacity. The other two operating policies do not abstract water from the Vassilika boreholes.

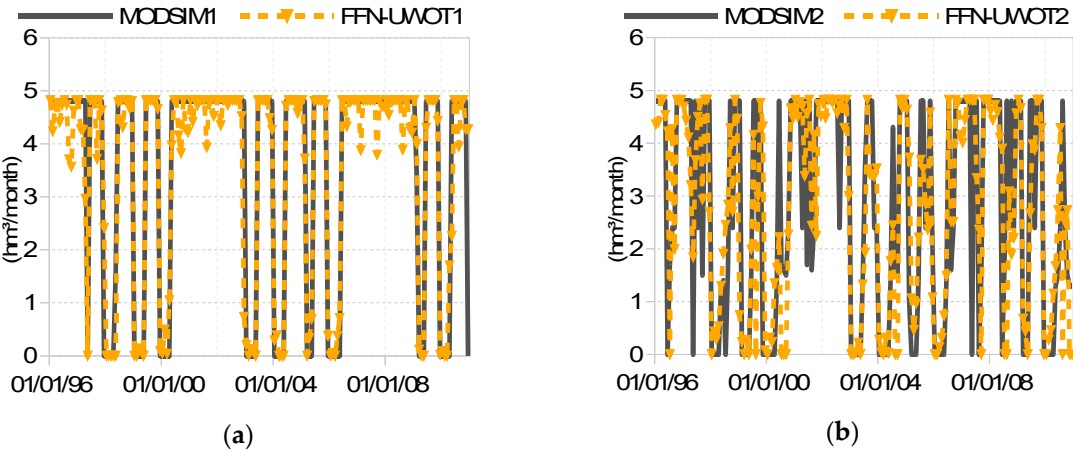

**Figure 6.** Groundwater abstractions from Vassilika obtained with MODSIM, and FFN with UWOT for operating policies No. 1 (**a**) and 2 (**b**).

Figure 7 displays the fluctuation of the storage in the Mornos reservoir for the 4 operating policies during the historical period. The simulated minimum storage level during this period gradually gets lower when going from operating policy No. 1 to operating policy No. 4.

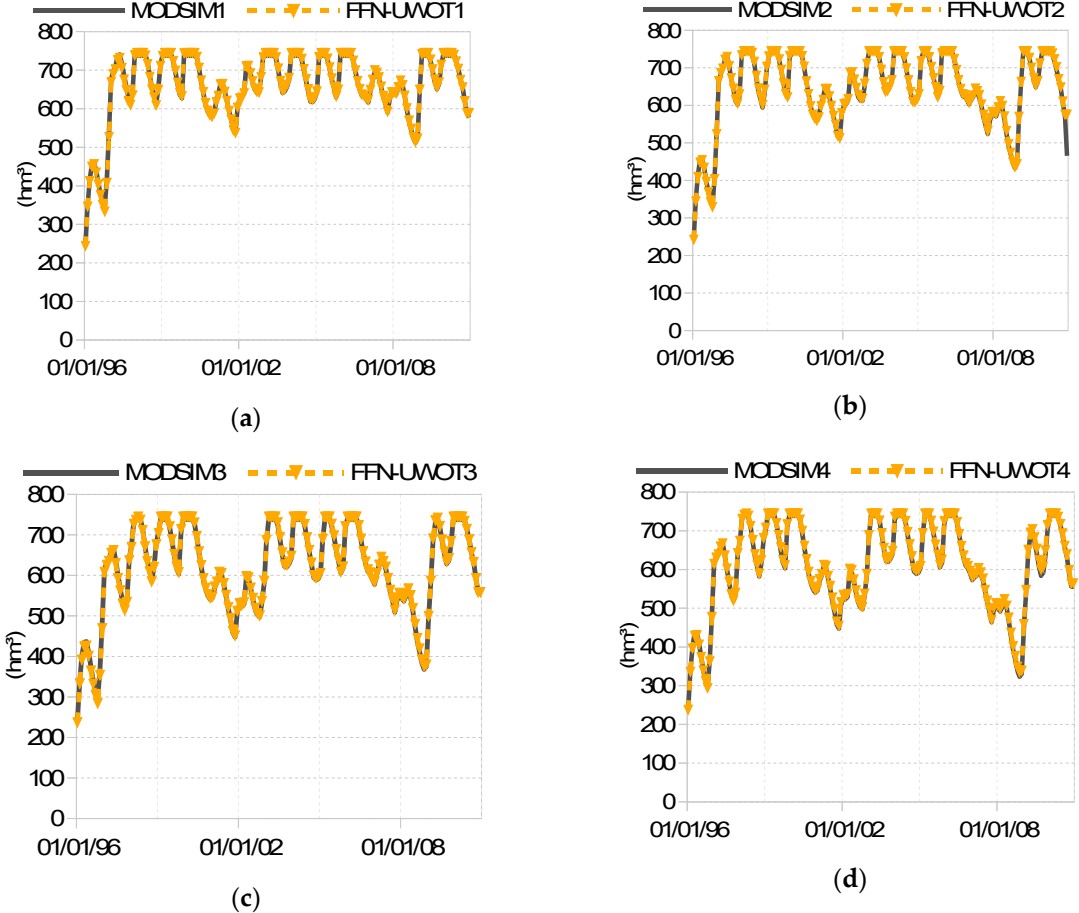

**Figure 7.** Storage in the Mornos reservoir obtained with MODSIM, and FFN with UWOT for operating policies No. 1 (**a**), No. 2 (**b**), No. 3 (**c**), and No. 4 (**d**).

Figure 8 displays the storage fluctuation in the Hyliki and Marathon reservoirs for operating policy No. 1. The other operating policies exhibit a similar pattern.

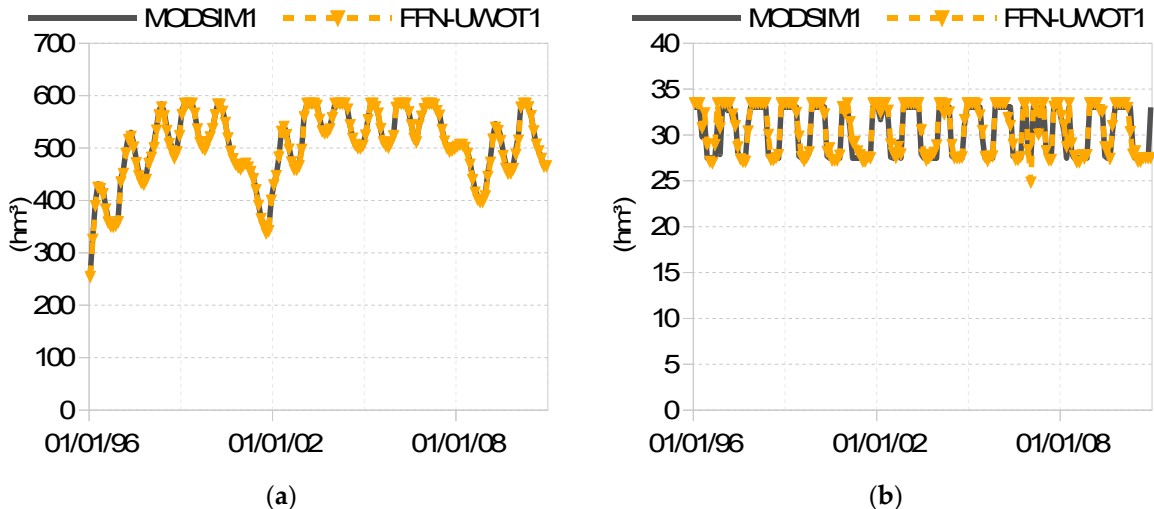

**Figure 8.** Storage in Hylilki (**a**) and Marathon (**b**) reservoirs obtained with MODSIM, and FFN with UWOT.

Figure 9 displays the trade-off between the energy consumed for water pumping and the minimum system storage (given as a percentage of the system total capacity) during the historical period and for operating policies No. 1 to 4.

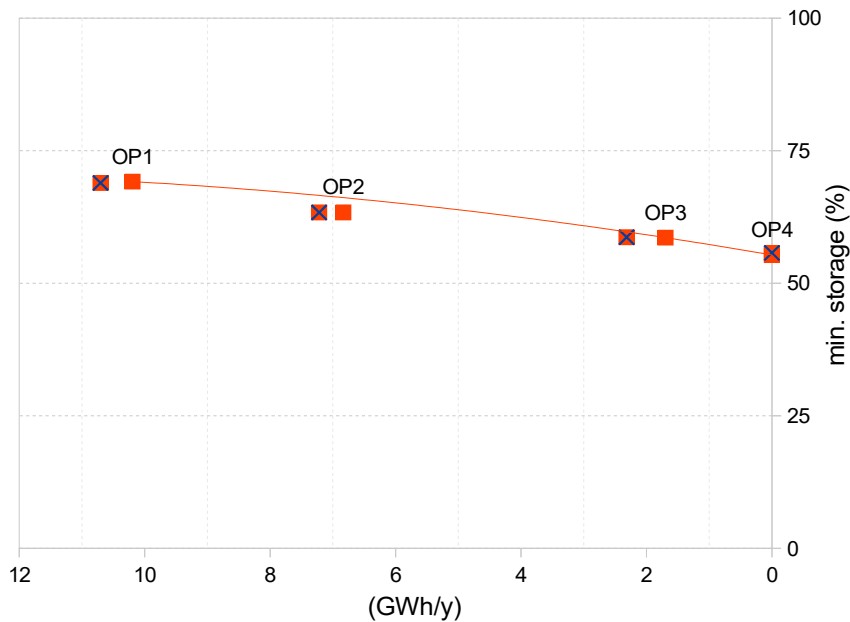

**Figure 9.** Energy/storage trade-off of operating policies No. 1 to 4. The square markers indicate the solutions obtained by MODSIM and the markers with a cross indicate the solutions obtained by the FFN. The red line is the trendline.

## 5. Discussion

The performance indicators provided in Table 2, and Figures 4–9 suggest that the FFN can successfully learn to operate the water supply system optimally (as it was instructed by the NFP model), complying with the guidelines of the operating policy (defined by the decision makers). It is worth noting that Equation (2) alone suffice to obtain the optimum system operation for any combination of storages/inflows/demands. Feeding this triplet into Equation (2) results in the ratio of flows between aqueducts that join at the nodes of the system. Then, the optimal abstractions from the available water resources can be obtained by multiplying the water demand by the appropriate ratios on the path between the demand and each of the resources.

Regarding the case study, the 4 operating policies resulted in 4 different operating rules (practically Equation (2) with 4 sets of weights and biases). All operating rules attempt to maximize abstractions from Marathon reservoir, which is a non-critical energy-free resource. Also, all operating rules avoid Hyliki, because is the second most energy-intensive resource after Mavrosouvala boreholes (the specific energies in kWh/m$^3$ of the energy-intensive resources can be found in the first row of Table 1, which is also the penalty values of operating policy No. 1). Mavrosouvala remains inactive in all 4 operating policies. Systematic usage of the Vassilika boreholes is suggested by operating rules No. 1 and 2, whereas the usage of this resource is not suggested by the other two rules. The contribution of the energy-intensive resources in each operating policy is summarized in Table 3.

**Table 3.** Use of energy-intensive resources.

| Operating Policy | Hyliki Abstractions | Vassilika Abstractions | Mavrosouvala Abstractions |
| :---: | :---: | :---: | :---: |
| 1 | ✓ | ✓ | – |
| 2 | – | ✓ | – |
| 3 | ✓ | – | – |
| 4 | – | – | – |

Figure 4 shows that the most prudent operating policy No. 1, almost eliminates the abstractions from Mornos reservoir during the dry period, to save the storage of this critical resource and hence, minimize the risk of failure in case of a prolonged drought. To achieve this, the operating rule of this policy maximizes the abstractions from the Hyliki reservoir and the Vassilika boreholes. The average amount of volume abstracted from the Mornos reservoir is lowest in operating policy No. 1 and highest in No. 4. More specifically, the average annually abstracted volumes are 289, 299, 325, 328 hm$^3$ respectively. These gradually increasing abstractions result in the gradually lower storage levels during the dry period when going from operating policy No. 1 to operating policy No. 4 (see Figures 7 and 9). In all cases, the FFN reproduced with very good accuracy the abstractions suggested by the optimization of the water supply system with MODSIM. It should be noted this optimization was not included in the FFN training, which was based exclusively on the simulations with synthetic data.

Figure 5 shows that the FFN reproduced also with very good accuracy the abstractions from the Hyliki reservoir, including the characteristic avoidance of abstractions for the majority of the time.

Figure 6 shows that the FFN reproduced with very good accuracy the abstractions from Vassilika boreholes in case of operating policy No. 1, whereas the accuracy was satisfactory for operating policy No. 2, in which Vassilika boreholes abstractions exhibit a much more complex pattern.

The fluctuation of the storage in the reservoirs is shown in Figures 7 and 8. The fluctuation of the storage in the Hyliki and Marathon reservoirs is similar across all four operating policies. The combination of FFN with UWOT simulated accurately the storage fluctuation in all reservoirs for all operating policies. The overall accuracy of the FFN with UWOT can be assessed in Figure 9. This figure gives the trade-off between energy consumption and the preservation of the Mornos reservoir storage achieved by MODSIM, and FFN with UWOT. The solutions of FFN with UWOT lay very close to the corresponding solutions of MODSIM for all operating policies, indicating the similarity of the two simulations.

The suggested methodology is generic and can be applied to any water supply system that can be optimized with NFP tools. Finally, the time required an Intel i7-7500U 3.5 GHz machine for the application of the MODSIM for 12,000 time steps was 30 s, whereas the FFN training took 2 min. The time required for the application of the FFN is negligible.

## 6. Conclusions

In this paper, the potential benefits of applying a machine learning algorithm in managing a water supply system were assessed. More specifically, a feedforward neural network (FFN) was trained utilizing the results of a network flow programming (NFP) model, which optimizes and simulates the operation of a water supply system. The penalty functions of the NFP were appropriately selected to reflect the operating policies with different levels of risk acceptance. The network flow programming was applied with synthetic data of a significant length to reliably capture the risk of each operating policy and provide a long training period for the FFN. The final product of this methodology is a trained FFN that can be represented as an algebraic formula. The demand-oriented approach employed renders this algebraic formula an operating rule that describes not only the desirable (the optimal) state of the system but also how to operate the entire system optimally, based only on the latest inflows and storages. To test the efficiency of this approach, a trained FFN was applied to a simplified version of the Athens water supply system. The results indicated that FFNs may be used reliably for supporting the operation of water supply systems.

The main advantages of the suggested methodology are the following:

1.  The optimization is very fast and accounts for the long-term persistence of the system stresses (i.e., '*the tendency of annual average streamflows to stay above or below their mean value for long periods*')
2.  Enabled by the demand-oriented approach, the suggested methodology does not require any specialized software for the decision support model. The 'how-to-operate' instructions via a simple algebraic formula can moderate the required level of expertise to use a decision support

tool because the algebraic formula can be easily wrapped in a user-friendly interface (a simple desktop application, a web-based application, a GIS-based application, even a spreadsheet).

Finally, it should be noted that the suggested methodology facilitates the integration of decision support tools with other kinds of models owing to its simplicity as well as speed of execution. The latter is essential in cases where multiple runs are required (model calibration, ensemble forecasting, etc.). For example, a hydrological model, a demand management model, and an FFN can be combined to simulate the water supply system along with the involved hydrological processes and social drives.

**Funding:** This research received no external funding.

**Conflicts of Interest:** The author declares no conflict of interest.

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
