# Peer review of "Machine Learning, Urban Water Resources Management and Operating Policy"

_resources, doi:10.3390/resources8040173_

Round 1

Reviewer 1 Report

The paper is well written with coherent content throughout. Just one minor suggestion here:

There are quite a few of neural networks that are suitable for this study, rather than accepting FFN straight away, a trial on some other networks are suggested to give reader reader on overview on how machine learning can help solve this problem. 

Author Response

The reviewer’s positive comments are much appreciated. Indeed, the field of machine learning offers a great variety of tools. In fact, the method of reinforcement learning was used initially in this study. After trying for one month and a half, other options were examined. The reason was that reinforcement learning was very slow to train and proposed a system operation that was not as good as that proposed by the classical methods. Some of the reasons for this poor performance are discussed by a Software Engineer at Google (Irpan, 2018). A comparison of the alternative methods that could be used in this problem would be of great scientific interest. However, considering the time that took to test only a single method (the reinforcement learning), this is something beyond the scope of this paper.

Irpan, A. 2018. Deep Reinforcement Learning Doesn't Work Yet, available online at: https://www.alexirpan.com/2018/02/14/rl-hard.html .

Reviewer 2 Report

The paper "Machine Learning, Urban Water Resources Management and Operating Policy" deals with the problems of the hydrogeological system regarding to the water supply of Athens. The notion of uncertainty is very present when describing certain parameters in hydrogeology. In this case, the author deals with the analysis of the inflows of this complex aquifers.

Author used the following scientific methodology to treat uncertainty: Bayesian decision theory, dynamic programming optimization method and stochastic modelling. On the other hand, the ability to using an artificial neural network (FFN) for successfully operate the system in accordance with a predefined operatng policy is shown.

Title: “Machine Learning, Urban Water Resources Management and Operating Policy“

Suggestion: It is proposed to change the title "Machine Learning, Urban Water Resources Management and Operating Policy" to " Machine Learning, Urban Water Resources Management and Operating Policy: Case study of the water supply system of Athens"

 Section "Introduction" (1): The problem of water resources management is presented. An overview of some references is presented.

Suggestion: It is proposed to transfer the Fig. 1 (with accompanying text) in Section “Methods”. The schema is a representation of the whole operating algorithm. Instead representation of the suggested methodology (text and figure 1) it is proposed to add at the end of the "Introduction" a short paragraph about how the paper is structured. These sentences help the reader to have a clear "map" while reading and to know what to expect (according to the established scheme: Introduction-Methodology-Application of the methodology on real case study-Results-Discussion-Conclusion).

Section "Materials and Methods" (2): This chapter provides an overview of applied methodology.

Suggestion: In general (paper), the problem-solving is detailed but it would be sufficient to show some more information about water supply system of Athens. It is proposed to create a new chapter (titled Case Study) in which the basic characteristics of the water supply system of Athens would be described - for example the spatial position of the aquifers (hydrogeological map or cross-section, GW source yield, hydraulic parameters ...) and any other information related to the city's water supply.

Section "Results" (3) and Section "Discussion" (4): The results are presented in a good way.

Suggestion: It is proposed (Fig. 4 - 8) to address the axes - add parameters on either side of the coordinate axes and put the units; it is suggested not to describe the parameters in the title of Figures. Figure 8 appears twice.

Section "Conclusions" (5): Please, explain the academic and scientific contribution of the paper. The conclusion is general.

Author Response

The author wishes to thank the reviewer for his constructive comments.

Regarding his comment to change the title, it is thought that though this would make more clear the location where the suggested methodology was tested, it would create the impression that this is a site-specific case study. This could hide the fact that the suggested methodology is generic, can be applied to any water supply system, and can be easily integrated with other models.

Regarding his comments on the Introduction section, this section was restructured. The figure and the description of the steps of the methodology were moved to the section Materials and methods. A paragraph to the end of Introduction that describes the structure of the manuscript was not added because the very standard scientific structure has been followed (Introduction, Materials and methods, Case study, Discussion, Conclusions).

A separate section that describes the case study has been added to the revised version of the manuscript.

Units have been added to the vertical axes of all figures. Two successive figures were having the caption number 8. This has been corrected.

The Conclusions section was rewritten to recapitulate more clearly the contribution of this study.

Reviewer 3 Report

The article concerns the optimization results for training a feedforward neural network. It is well-organized. Line 127: Figure 1: Schematic representation of the suggested methodology, the Figure 1 should be presented before the advantage of the suggested methodology. The title of the manuscript is not suitable, it should be changed, the title should accurately reflect the paper's content. Page 4: Line 138: How you define acceptable risk? The conclusions should be justified by the published data. In the conclusions some obtained results should be interpreted and deliver the meanings, this section should be more focused and based on the obtained results.

Author Response

The author wishes to thank the reviewer for his constructive comments.

The Introduction section has been restructured. Figure 1 has been moved to the next section (Materials and methods).

This study presents how a feedforward neural network should be trained to manage a water supply system optimally, i.e., to minimize the operational cost given the acceptable risk level. The latter is included in the policy decisions the water utility company should make. Policy decisions govern future policy and operations decisions. Therefore, this study suggests an approach employing machine learning that, given a policy decision, suggests the optimum management of the urban water resources. The title contains descriptive words that associate strongly with the content of the manuscript. For this reason, the author feels that the title should remain as it is.

The acceptable risk is defined by the water utility company. The following text has been added: “This level depends on the acceptable risk, a policy decision made by the water utility company.

The Conclusions section was rewritten to recapitulate more clearly the contribution of this study.

Reviewer 4 Report

The paper ‘Machine Learning, Urban Water Resources Management and Operating Policy’ is interesting and well-written. It deals mainly with the development of a FFN network serving as a decision support tool on the functioning of a water supply system, particularly as far as the reservoir operation is concerned (but not limited to it). I suggest performing minor improvements to the manuscript, as detailed in the following.

Abstract. The main issue is that it is difficult to precisely identify the topic the authors are dealing with. There is a general reference to ‘water services’ but this is not enough to clarify the focus of the activity. Are you referring to water allocation in case of scarcity?

Introduction.

Lines 29-33. Please add some references. Following my previous comment, the topic is still too broad. Specific reference should be made to the main objective(s) of the proposed model (e.g. optimization of allocation for a multi-reservoir system?) Line 124. Reference is made to the stakeholders. Which stakeholders could be potentially interested by (or users) the proposed tools? Lines 125-126. Too vague, please rephrase.

Materials and methods

This section should provide general information on the methodology. I would not include information on the case study, which should be included in a separate section. Lines 160-166. Concerning the historical data, it is not clear whether you are referring to the inflows only, or to the variations in demand, use, population, etc. over the years. Line 167. How is the objective function structured? Please provide the equation

Discussion

What is the potential replicability of the approach? Which are the limitations to its use in other situations and case studies?

Author Response

The author wishes to thank the reviewer for his constructive comments. The following text describes how his comments were taken into account.

The vague term “water services” has been replaced with “the management of the urban water resources”.

The following references were added to the first paragraph of the Introduction section to make more clear the focus of this study.

1. Philbrick, C.R.; Kitanidis, P. Limitations of deterministic optimization applied to reservoir operations, Journal of Water Resources Planning and Management, 1999, ASCE, 125, 135–142, https://doi.org/10.1061/(ASCE)0733-9496(1999)125:3(135).

2. Grigg, N.S. Water Resources Management, McGraw-Hill, New York, 1996.

The text of line 124 (now line 88) was changed to “For example, the tool can be implemented as a spreadsheet or a GIS based application that helps a utility manager to plan the operation of a water supply system.”

The sentence in lines 125-126 (now lines 90-91) was changed to “Furthermore, researchers can use this approach to achieve any level of model coupling [16] of decision support models with other types of models.”

The Introduction, and Materials and Methods sections have been restructured. A new section has been added that includes a description of the case study.

The following text has been added to clarify the type of historical time-series “(time-series of inflows and water supplied to Athens)”.

The objective function is the standard equation used in network flow programming. A reference to the MODSIM manual has been added, where this equation is described with more details.

The following text was added to the Discussion section to make clear the limitations of the suggested methodology: “The suggested methodology is generic and can be applied to any water supply system that can be optimized with NFP tools.”

Round 2

Reviewer 2 Report

The author has made a lot of effort to improve the quality of work.

The author partially accepted the suggestions.

It may be better to use "paper" instead of "study" (Section Conclusion).

There is no need to cite in the conclusion (Ref. 20).

Author Response

The author wishes to thank the reviewer for his suggestions and comments.

The word “study” has been replaced with “paper”.

The reference has been removed from the Conclusions section.

Reviewer 3 Report

The article deals with the urban water resources management and operating policy using machine learning, it is presented in the city scale in Athens. The following suggestions should be reffered to. Line 459. the choice of reference should distinguish the importance of the concept of the tolerable water shortage level and criteria in terms of risk acceptance, as presented in Rak,J.; Pietrucha-Urbanik, K. An approach to determine risk indices for drinking water – study investigation. Sustainability-Basel, 2019, 11, 3189. The last point of the article contains in fact only the conclusions relating to the researched case study, with the perspective concerning hydrological processes and social drives, but there is no more detailed perspective. Besides, there is no discussion about possible limitations of using the proposed modelling. Therefore, I propose to consider the possibility of completing the last point for discussion on limitations of the use of such analysis.

Author Response

Every effort was made to address the reviewer’s suggestions.

There is no line 459 in the manuscript. The last line of the manuscript is 415.

The study indicated by the reviewer investigates the quality-reliability. There is a fundamental difference between the reliability regarding quality and quantity. The latter involves a binary assessment of the provided service, whereas the former involves a linear assessment (see Exceedance Index). For this reason, the annual probability of failure of the water supply system is given by equation (1). The value calculated by (1) should not exceed the acceptable risk set by the water utility company.

The last point of the article (the one mentioning hydrological processes and social drives) provides an idea for further research. The main advantages of the suggested methodology are clearly stated in bullets 1 and 2 of the Conclusions section.

The limitations of the suggested methodology are mentioned in the Discussion section: “The suggested methodology is generic and can be applied to any water supply system that can be optimized with NFP tools.”